# Preparation of Affinity Purified Antibodies against ε-Glutaryl-Lysine Residues in Proteins for Investigation of Glutarylated Proteins in Animal Tissues

**DOI:** 10.3390/biom11081168

**Published:** 2021-08-07

**Authors:** Artem V. Artiukhov, Ekaterina F. Kolesanova, Aleksandra I. Boyko, Anastasiya A. Chashnikova, Sergei N. Gnedoy, Thilo Kaehne, Daria A. Ivanova, Alyona V. Kolesnichenko, Vasily A. Aleshin, Victoria I. Bunik

**Affiliations:** 1Faculty of Bioengineering and Bioinformatics, Lomonosov Moscow State University, 119991 Moscow, Russia; whitelord_1994@mail.ru (A.V.A.); boiko.sash@gmail.com (A.I.B.); vbunik@yahoo.com (A.A.C.); aleshin_vasily@mail.ru (V.A.A.); 2Belozersky Institute of Physico-Chemical Biology, Lomonosov Moscow State University, 119991 Moscow, Russia; 3Laboratory of Peptide Engineering, Institute of Biomedical Chemistry, 10/8, Pogodinskaya ul., 119121 Moscow, Russia; EKolesanova@yandex.ru (D.A.I.); alenka.aks@mail.ru (A.V.K.); 4“Agrobiomed” Ltd., 249010 Borovsk, Russia; sngnedoy@mail.ru; 5Institute of Experimental Internal Medicine, Otto-von-Guericke University, 39120 Magdeburg, Germany; thilo.kaehne@med.ovgu.de; 6Department of Biological Chemistry, Sechenov First Moscow State Medical University, 119146 Moscow, Russia

**Keywords:** antibodies against ε-glutaryl-lysine residues, affinity chromatography, glutarylation, succinylation, acetylation

## Abstract

The glutarylation of lysine residues in proteins attracts attention as a possible mechanism of metabolic regulation, perturbed in pathologies. The visualization of protein glutarylation by antibodies specific to ε-glutaryl-lysine residues may be particularly useful to reveal pathogenic mutations in the relevant enzymes. We purified such antibodies from the rabbit antiserum, obtained after sequential immunization with two artificially glutarylated proteins, using affinity chromatography on ε-glutaryl-lysine-containing sorbents. Employing these anti(ε-glutaryl-lysine)-antibodies for the immunoblotting analysis of rat tissues and mitochondria has demonstrated the sample-specific patterns of protein glutarylation. The study of the protein glutarylation in rat tissue homogenates revealed a time-dependent fragmentation of glutarylated proteins in these preparations. The process may complicate the investigation of potential changes in the acylation level of specific protein bands when studying time-dependent effects of the acylation regulators. In the rat brain, the protein glutarylation, succinylation and acetylation patterns obtained upon the immunoblotting of the same sample with the corresponding antibodies are shown to differ. Specific combinations of molecular masses of major protein bands in the different acylation patterns confirm the selectivity of the anti(ε-glutaryl-lysine)-antibodies obtained in this work. Hence, our affinity-purified anti(ε-glutaryllysine)-antibodies provide an effective tool to characterize protein glutarylation, revealing its specific pattern, compared to acetylation and succinylation, in complex protein mixtures.

## 1. Introduction

Protein glutarylation is a relatively recently discovered post-translational modification of lysine residues, observed from prokaryotes (*E. coli*, *M. tuberculosis*) to eukaryotes [1,2,3,4,5]. Glutarylation occurs in proteins of various cellular and extracellular compartments, such as mitochondria, cytoplasm, cell nucleus and blood serum [1,2,3,6,7,8,9,10,11]. Similar to succinylation and malonylation, the glutarylation of protein lysine residues not only increases the length of the lysine side chain, but also changes the charge of the residue from “+” to “−”, which essentially discriminates these types of acylations from the most known lysine acetylation. The ensuing reorganization of salt bridges that exist between unmodified lysine residues and dicarboxylic amino acid residues may affect the protein conformation and/or binding of negatively charged substrates, coenzymes, effectors and DNA [3,6,11,12,13]. In most proteins, glutarylation is less abundant than acetylation or succinylation, although in some proteins or under specific conditions, a significant number of lysine residues may be glutarylated [1,6,7,11,12]. The latter refers mainly to proteins involved in the metabolism of amino acids, carboxylic acids and to histones [1,6,7,8,9,11,12,13,14]. The transient nature and organ-specificity of protein glutarylation [1,6,10], potential competition between glutarylation and other modifications of lysine residues in proteins (acetylation, succinylation) [5,6,7,8,14,15] suggest glutarylation to be an important component of metabolic regulation. In particular, glutarylation of histones is involved in the regulation of gene expression [3,9,11,12,13]. The glutarylation of enzymes of tricarboxylic acid and urea cycles, of the amino acids and fatty acids metabolism is known to regulate enzymatic activities, exemplified by glutamate dehydrogenase (GDH, EC: 1.4.1.2), isocitrate dehydrogenase (EC: 1.1.1.-) and carbamoyl phosphate synthetase 1 (EC: 6.3.4.16) [1,4,5,6,7,8,13]. Changed glutarylation of these metabolic enzymes and other proteins may be involved in mechanisms of pathological processes [6,7,8,9,13,16,17,18,19]. For instance, the glutarylation of serum albumin (presumably occurring in the liver) is significantly reduced in patients with acute myocardial infarction, which may mark a decreased metabolic flux in the liver of such patients [9]. A decrease in the glutarylation of a number of sperm proteins negatively affects the mobility of these germ cells, which is restored upon the artificial stimulation of glutarylation [10]. The deglutarylation of isocitrate dehydrogenase and glucose-6-phosphate dehydrogenase, leading to the restoration of the activity of these enzymes, is an important factor in the cellular response to oxidative stress [7]. An increased glutarylation of the brain mitochondrial enzymes, particularly GDH, during the accumulation of glutaryl-CoA and glutaric acid in the body, or upon blocking the production of sirtuin 5 deacylase, leads to neurological disorders [6,8,13,18]. Nevertheless, the glutarylation of lysine residues in proteins and its role in normal and pathologically altered processes of cellular metabolism are insufficiently studied.

Antibodies specific to the ε-N-glutaryl-lysine (ε-glutaryl-lysine) residue are an important tool for the study of protein glutarylation. Such antibodies are used both to establish a general pattern of protein glutarylation using immunoblotting and immunohistochemical methods [1,6,7,9,10], and to extract glutarylated protein fragments after proteolysis for mass spectrometric detection and analysis of the extracted peptides [2,6,9]. A single preparation of polyclonal anti-(ε-glutaryl-lysine) antibodies is available from two biotechnological companies, provided as a reagent for immunoblotting (PTM Biolabs, PTM-1151) or linked to beads for glutarylated peptide extraction (Cell Signaling, #26101). However, affinity purified preparations of polyclonal antibodies obtained by researchers themselves are often employed in large-scale studies [1,3,6]. This procedure is also useful when antibodies to specific peptides are desired. Notably, the use of diverse glutarylated proteins as antigens, and specific affinity purification procedures [1,6], may result in the production of anti-(ε-glutaryl-lysine) antibodies that may slightly differ in their fine specificities; nevertheless, they should retain selectivity to ε-glutaryl-lysine residues and be capable of recognizing glutarylated proteins. The goal of our current work is to obtain antibodies that specifically interact with ε-glutaryl-lysine residues for the determination of glutarylated proteins in animal tissues. An overall assessment of protein glutarylation in tissue homogenates and/or subcellular fractions may rapidly detect certain metabolic and genetic disorders via an efficient and minimally invasive diagnostic procedure [9,10]. Thus, the anti(ε-glutaryl-lysine) antibodies may be applied for both scientific and diagnostic purposes.

## 2. Materials and Methods

***Chemical modification of lysine residues of proteins with glutaric anhydride.*** GDH from bovine liver (“Sigma-Aldrich”, St. Louis, MO, USA) was acylated with glutaric anhydride via incubating the protein (1.84 mg/mL) in the presence of the anhydride (0.16 mM) in 0.1 M Tris-HCl buffer, pH 7.5 on ice for 15 min, according to the procedure described before [20]. BSA (“Boval BioSolutions-Proluant Biologicals”, IA, USA) and malate dehydrogenase (MDH, total preparation of MDH1 and MDH2, “REANAL”, Budapest, Hungary) glutarylations were carried out in a similar manner. Separation of the excess reagent from the protein was not required, since the unreacted anhydride was completely hydrolyzed [20]. The same procedure was employed for acetylation and succinylation of the purified proteins (GDH and BSA) using acetic and succinic anhydrides, correspondingly.

***Mass spectrometric analysis of glutarylated proteins.*** The modified and unmodified GDH preparations were subjected for SDS-PAGE disc-electrophoresis (the concentration of the separating gel was 10%) [21]. Gel bands corresponding either to unmodified or glutarylated GDH were excised, subjected to proteolysis by trypsin, and the resulting peptide fragments were analyzed by HPLC with mass-spectrometric detection of modified and unmodified peptides according to the previously described protocol [22]. The relative level of K503 GDH glutarylation was calculated in arbitrary units (a.u.), as the ratio of peak areas of peptides containing glutarylated K503 to peak areas of the corresponding unmodified GDH peptides, according to the previously described method [22].

***Production of polyclonal antibodies against glutarylated proteins.*** Antiserum against glutarylated proteins was obtained by immunization of male Californian rabbits (4–4.5 kg). Antigen solutions in sterile saline were mixed with equal volumes of complete or incomplete Freund’s adjuvant (“Difco Laboratories”, Detroit, MI, USA). A dose of 250 µg antigen per 1 animal in 1 mL of the emulsion was injected intradermally in at least 20 points of the neck skin wrinkle. The first cycle of the immunization was performed with glutaryl-GDH as an antigen. Glutaryl-BSA was used as an antigen for the second cycle, 2 months after the first one. Scheme of the immunization was common for the two cycles: 1st immunization—protein with complete FA; 2nd immunization, two weeks later—protein with incomplete FA; 3rd immunization—the same as the 2nd one; 4th immunization, one week later—protein with incomplete FA; 5th immunization—the same as the 4th one. Rabbits were bled through ear veins one week after the last immunization. Sera were obtained after clotting via centrifugation at 10,000 rpm and kept at +4 °C with 0.05% NaN_3_.

Anti-glutaryl-GDH or glutaryl-BSA antibody titers were determined in serum by ELISA in 96-well polystyrene high binding plates (“Corning”, Corning, NY, USA). Proteins (0.5 μg/well) were adsorbed onto well bottoms from phosphate-buffered saline, pH 7.2 (PBS). Casein hydrolysate for parenteral injections (“Microgen”, Makhachkala, Russia) diluted 1:5 (v/v) with PBS was used as blocking buffer. Horseradish peroxidase-conjugated goat antibodies against rabbit IgG (“Helicon”, Moscow, Russia) were employed as secondary antibodies. Bound antibodies were visualized with peroxidase substrate and cosubstrate solution (0.015% (w/v) H_2_O_2_, 132 μg/mL tetramethylbenzidine (TMB, “Imtek”, Moscow, Russia) in 0.1 M sodium citrate buffer, pH 4.9 with 1% dimethylsulfoxide). TMB was dissolved in DMSO at a concentration of 13.2 mg/mL and 100 μL of this solution was added to 10 mL buffered substrate solution. Optical density was measured at 450 nm, plate reader “UNIPLAN” (“PIKON” CAS, Moscow, Russia). Rabbit antiserum obtained before the immunizations, was used as a control.

***Synthesis of affinity sorbents for the purification of antibodies against glutarylated proteins.*** Ligands for affinity purification of antibodies against glutarylated proteins—ε-glutaryl-Lys-Gly-amide (ligand 1) and Gly-Lys(ε-glutaryl)-amide (ligand 2)—were prepared by solid-phase synthesis on Rink Amide polystyrene matrix (Rink Amide AM, “ChemPep”, Wellington, FL, USA) using 9-fluorenyl(methoxycarbonyl)(Fmoc)-protected amino acids (Fmoc-Lys(4-methyltrityl) and Fmoc-Gly, «ChemPep») and di(*iso*-propyl)carbodiimide activation in N,N-dimethylformamide (DMF) [23]. After removing the 4-methyltrityl group with 2% trifluoroacetic acid (TFA) in dichloromethane [24], Lys residue was glutarylated on the matrix with glutaric anhydride in DMF in the presence of N-ethyl-N,N-di-*iso*-propylamine. Fmoc-deprotection was performed with the solution of 2% 4-methylpiperidine and 2% 1,8-diazabicyclo [4,5,0] undec-7-ene in DMF [23]. Peptides were cleaved from the matrix with a TFA-water (19:1, v/v) mixture and precipitated with ether. Ligands were further purified by semi-preparative reversed-phase HPLC, column YMC TRIART C18, 21 × 250 mm, 10 mkm (“YMC Europe”, Dinslaken, Germany), workstation Agilent 1100 (“Agilent Technologies”, Santa Clara, CA, USA). Analytical reversed-phase HPLC with mass-spectral detection showed at least 95% ligand purity.

Affinity sorbents for specific antibody purification were prepared by Affi-Gel 15 (“Bio-Rad Laboratories”, Hercules, CA, USA) modification with ligands according to the producer manual. An amount of 13 mg of either ligand 1 or 2 (24 mmol each) were dissolved in 2 mL 0.1 M Na-Hepes buffer, 0.2 M NaCl, pH 7.5. Ice-cold ligand solutions were added each to a 1.6 mL portion of Affi-Gel 15 (preliminarily washed with the same buffer), mixed thoroughly and incubated 4 h at 4 °C. Ligand binding to the matrix was controlled via the filtrate reactivity with 1-fluor-2,4-dinitrobenzene after the reaction completion. Unreacted active matrix groups were blocked with tris(hydroxymethyl)aminomethane (Tris) upon slow washing of each portion of the matrix with 16 mL 0.1 M Tris-HCl buffer, pH 7.5. After that, the modified matrix was washed with 16 mL 20 mM phosphate buffer, pH 7.2 (PB), 10 mL PB with 1 M NaCl, 10 mL 0.15 M glycine-HCl buffer, pH 2.4, containing 10% 1,2-propanediol and equilibrated with PB. All buffers used for affinity chromatography contained 0.025% (w/v) NaN_3_.

***Affinity purification of antibodies against glutarylated proteins***. IgG fractions from immunized rabbit sera were prepared by ammonium sulfate fractionation followed by separation from other proteins on DEAE Blue agarose (DEAE Blue Affi-gel, «Bio-Rad Laboratories») [25,26] in 20 mM PB, pH 7.2.

Rabbit IgG fraction containing antibodies against glutaryl-GDH and glutaryl-BSA was applied to the affinity sorbent. An amount of 4 mL IgG fractions with protein concentration 300 μg/mL were slowly (for about 2 h) applied onto 1.6 mL columns with Affi-Gel 15 modified with either ligand 1 or ligand 2. After that, the columns were washed with 10 mL PB, 10 mL PB with 1 M NaCl. Ligand-bound antibodies were eluted with 8 mL 0.15 M glycine-HCl buffer, pH 2.4, containing 10% 1,2-propanediol. pH of eluted fractions was brought to 7.5–7.8 with 1 M Tris-HCl, pH 8.7 immediately after the elution.

***Immunodot assay and immunoblotting.*** For immunodot analysis, 2 μL of glutarylated and non-glutarylated protein samples (GDH, MDH and BSA) at 1 mg/mL (GDH and BSA) or at 2 mg/mL (MDH) in Laemmli buffer (0.082 M Tris-HCl buffer, pH 6.8, 2% SDS, 10% glycerol) were loaded onto a nitrocellulose membrane (“Thermo Fisher Scientific”, Waltham, MA, USA). Non-specific binding was blocked with iBind Solution Kit (“Thermo Fisher Scientific”) for 30 min at room temperature. After washing in TBST buffer (20 mM Tris-HCl, pH 7.5, 0.14 M NaCl, and 0.2% Tween 20), the membranes were incubated with affinity-eluted antibody preparations obtained from antisera after the second cycle of immunization and diluted 1:5, 1:25, 1:125, 1:625 and 1:3125 in TBST buffer, for 1 h at 20 °C with constant stirring. Peroxidase-labeled goat antibodies against rabbit IgG (“Cell Signaling Technology”, Danvers, MA, USA, 1:2000 in TBST buffer) were used as anti-species antibodies. The bound antibodies were developed by incubating the membranes in Clarity Max Western ECL Substrate Kit (“Bio-Rad Laboratories”). Comparison of the home-made anti(ε-glutaryl-lysine) antibodies and commercial Pan anti-glutaryl-lysine rabbit polyclonal antibodies from “PTM Biolabs“ (Chicago, IL, USA) (PTM 1151) was performed by immunodot analysis with chemically glutarylated, acetylated, succinylated and unmodified proteins, using 0.92 μg protein per one dot. The modifications were described above.

Glutarylated proteins of the rat cerebral cortex or liver were determined via immunoblotting of homogenates of tissues of male rats and in extracts of mitochondrial fractions from these tissues prepared according to previously published protocols [27]. For immunoblotting, proteins were separated by SDS-PAGE according to Laemmli [21]. 2,2,2-Trichloroethanol was added to resolving gel containing 10% acrylamide for subsequent semi-quantitative determination of total protein by fluorescence under UV irradiation of the gel [28,29]. A total of 2 μg of the purified protein preparations or 22 μg of homogenate or mitochondrial extract total proteins was loaded onto the gel lanes. To study potential interconversion of the glutarylated bands resulting from endogenous proteolysis, incubation of the homogenates prior to the gel electrophoresis was performed at 37 °C for up to 3 h in a buffer, pH 8, containing 40 mM MOPS, 50 mM Tris-HCl, 0.15 mM EGTA, 4 mM MgCl_2_, 50 mM NaCl, 1 mM dithiothreitol, 20% glycerol, 0.75% Triton X-100 and protease inhibitors (0.15 mM AEBSF, 0.12 μM aprotinin, 2.5 μM bestatin, 2.25 μM E-64, 1.5 μM leupeptin and 1.05 μM pepstatin A).

After the electrophoresis, proteins were visualized in the gels by 2,2,2-trichloroethanol-enhanced fluorescence and transferred onto a polyvinylidene fluoride (PVDF) membrane by semi-dry electrotransfer using a Power Blotter Station (Thermo Fisher Scientific) and Power Blotter 1-Step Transfer Buffer (Thermo Fisher Scientific) for 10 min at 2.5 A, 25 V. Immunovisualization of glutarylated proteins was performed using affinity purified antibodies as described above. The dilutions used were indicated in the figure captions. Anti-succinyl-lysine (PTM Biolabs #PTM-401, 1:2000) and anti-acetyl-lysine (“Cell Signaling Technologies” #9814, 1:1000) antibodies were used in a comparative analysis of antibodies against different types of lysine residue acylations.

Gels and membranes were analyzed using ChemiDoc MP System (“Bio-Rad Laboratories”). Mixtures of prestained (Bio-Rad #1610374, 1:7) and unstained strep-tagged (Bio-Rad #1610374, 1:120) marker proteins with known molecular weights were used to estimate the molecular weights of protein bands in the gel and on the membrane. Semi-quantitative estimation of luminescence (for glutarylated proteins) or fluorescence (for total protein) intensity of the stained bands was performed using the Image Lab software (“Bio-Rad Laboratories”).

Protein concentrations in homogenates and mitochondrial extracts were determined by the Bradford method [30]; its micro variant [31] was used for the preparations of affinity purified antibodies.

## 3. Results and Discussion

### 3.1. Glutarylation of Proteins In Vitro

Commercial preparations of purified proteins (GDH from bovine liver, MDH from porcine heart and BSA) were modified with glutaric anhydride under the same conditions. The glutarylation under our conditions was confirmed by a comparative mass spectrometric analysis of peptide maps, using the samples of modified and unmodified GDH. A mass spectrometric analysis, carried out according to the published protocol [22], revealed a significant increase in the level of glutarylation of lysine residues in GDH upon treatment of the enzyme with glutaric anhydride. In the original GDH preparation, the degree of glutarylation of residues K171, K183, K187, K191, K390, K527 and K503 was very low compared to the modified protein. Treatment with glutaric anhydride led to a significant increase in the level of glutarylation of these and a number of other lysine residues in GDH. For example, the relative amount of the peptide-containing glutarylated lysine residue 503 (ISGASEKDIVHSGLAYTMER peptide; MS/MS spectra of glutarylated and non-glutarylated peptides are given in Appendix A) in the studied preparation of GDH, normalized to the sum of those for the well identifiable GDH peptides, not containing lysine residues (DSNYHLLMSVQESLER and DDGSWEVIEGYR) in the same preparation was increased by glutaric anhydride more than 10 times—from 0.002 ± 0.001 to 0.0320 ± 0.045 a.u. As noted earlier [22], these ratios were characterized by arbitrary units, which can be used for a comparative analysis of changes in the level of modification in the same enzyme preparation based on the determination of the same peptides. It should be kept in mind that such ratios do not characterize the percentage of the modified residue in the preparation, since the level of mass spectrometric identification of different peptides depends on many factors.

A comparison with published data [6] for a similar modification of GDH by a natural acylating agent, glutaryl-CoA, showed a noticeable, but incomplete, overlap of the enzyme lysine residues modified by both agents (Table 1). In overall, 9 and 12 peptides comprising the glutarylated lysine residues were determined after modification by 0.16 mM glutaric anhydride in this work and 0.5 mM glutaryl-CoA in an independent study [6]. Thus, the deeper modification was observed at the higher concentration of glutaryl-CoA compared to the lower concentration of glutaric anhydride. It is worth noting that the protein modification by glutaryl-CoA occurs through an intermediate formation of glutaric anhydride [32]. Hence, the observed differences in the modified peptides may manifest differences in the accessibilities and/or affinities of the residues in specific protein sites to the employed reagents. Moreover, the prolongated action of glutaric anhydride generated from glutaryl-CoA by intramolecular catalysis [32], compared to the less stable glutaric anhydride itself, may also contribute to the deeper modification in the former case compared to the latter one. Finally, an initial difference in the modification of specific residues in the two procedures may have caused different kinetics of GDH unfolding, exposing varied modifiable lysine residues during the reaction time.

### 3.2. Preparation and Characterization of Antibodies against Glutarylated Proteins

In our previous work, we used sequential immunization cycles with the replacement of the carrier protein at the same hapten to obtain anti-peptide antibodies that selectively interacted with one of the highly identical isoforms of the eukaryotic translation elongation factor 1A, 1A2 (eEF1A2) [26]. A similar approach was used to obtain antibodies to glutarylated proteins in the current work. For the first cycle of rabbit immunization, glutarylated GDH was employed and the second cycle of immunization was performed with the preparation of glutaryl-BSA. After the first cycle of immunization, a high antibody titer of 1:102,400 for glutaryl-GDH was determined, but the antibody titer for the non-glutarylated enzyme was just as high. The titer of antibodies against glutaryl-BSA in the rabbit antiserum obtained after the second cycle of immunization increased from 1:3200 (after the first cycle of immunization) to 1:102,400. At the same time, the titer of antibodies against glutaryl-GDH did not change and remained equal to 1:102,400, while the titer of antibodies against unmodified GDH decreased by 16 times. Thus, sequential cycles of immunization with two different glutarylated proteins increased the relative portion of the specific anti(ε-glutaryl-lysine) antibodies, not dependent of the carrier protein. Such a conclusion was confirmed below by results of the antibody affinity purification on the two different peptide ligands.

Two different affinity matrices with a lysine residue, glutarylated at its ε-amino-group, were prepared from Affi-Gel 15 for the affinity purification of the obtained antibodies. The first matrix comprised ligand one, ε-glutaryl-lysyl-glycinamide, and the second one comprised ligand two, glycyl-ε-glutaryl-lysinamide. Since lysine residues, which are subject to glutarylation, are usually located in the middle of peptide chains, the ligand amidation was used to mimic a peptide bond. The glycine residue played the same role; this residue was selected for its minimal size among amino acid residues. According to the results of the unbound ligand reactions with 1-fluoro-2,4-dinitrobenzene, the degree of conjugation of each ligand with the matrix was 14 micromoles per 1.6 mL of the matrix. Thus, the sorbents with ligands one and two differed in the location of the ε-glutaryl-lysine residue relative to the active carrier groups. Upon purification on columns of the same volume (1.6 mL), the ligand one-based column produced approximately twice as much IgG per chromatography cycle as the ligand-two-based column. Hence, the IgG yield from the ligand-one-based column was 26 µg IgG vs. An amount of 14 µg IgG from the ligand-one-based column. The fractions of antibodies not bound to these affinity matrices contained fairly large amounts of antibodies interacting with both glutarylated and non-glutarylated proteins (data not shown). The antibodies obtained by affinity purification on both the ligand-one- and ligand-two-incorporating matrices demonstrated selectivity for glutarylated proteins compared to the unmodified ones. Figure 1 compares the reactions of IgGs purified by affinity chromatography on matrices comprising ligand one or ligand two, from the antiserum obtained after the two immunization cycles, with glutarylated and non-glutarylated GDH, MDH and BSA. No significant differences in the selectivity of both types of antibodies against glutarylated proteins in the range of optimal concentrations for immunoblotting (0.01–0.13 µg/mL) were obvious. In further experiments, we used antibodies purified on the ligand-one-comprising matrix, which was characterized by a higher IgG yield.

Thus, highly selective antibodies towards glutarylated lysine residues were obtained after the two immunization cycles (Figure 1). Some reactivity observed at the high concentrations of antibodies against unmodified commercial GDH and BSA preparations may have resulted from a low level of the known physiological glutarylation of the enzyme and a large amount of the purified protein used for this assay. As indicated above, we identified a low level of glutarylation of a number of lysine residues in the non-treated preparation of purified liver GDH, which was also observed by others [6]. Some glutarylation of the commercial BSA preparations was likely in view of the known glutarylation of human serum albumin [9], although this modification was not detectable in BSA in an independent study [32]. So, the antibody interaction with unmodified BSA may have been caused by either its very low glutarylation level, or unspecific interaction. Nevertheless, Figure 1 demonstrates a minor contribution of this immunostaining, compared to that of the glutarylated BSA, especially under the optimal concentration of the antibodies and the affinity purification employed on ligand one. MDH glutarylation in the glutaric-anhydride-treated samples was the least pronounced. Thus, the choice of ε-glutaryl-lysine residues within the peptide backbone as affinity ligands was successful for the purification of antibodies that specifically interacted with glutarylated proteins.

It should be noted that glycine residues was adjacent to 4 out of 32 lysine residues in bovine GDH, 4 out of 60 lysine residues in BSA, 2 out of 31 lysine residues in porcine cytoplasmic MDH and 9 out of 25 lysine residues in porcine mitochondrial MDH. Taking into account the lowest reactivity of glutarylated MDH out of the three studied proteins to the antibodies (Figure 1) and the fact that only one of the nine mass-spectrometry-identified ε-glutarylated GDH lysine residues was adjacent to the glycine residue (K480, Table 1), the interaction of the obtained antibodies with glutarylated proteins was determined solely by the ε-glutarylation of their lysine residues, and not by the proximity of ε-glutaryl-lysine to glycine residues realized in the affinity ligand.

It is worth noting that increasing concentrations of interacting partners, such as the glutarylated proteins and anti(ε-glutaryl-lysine) antibodies, inevitably increases non-specific interactions, which may contribute to the observed minor interaction of the affinity-purified anti(ε-glutaryl-lysine) antibodies with unmodified BSA and MDH at a high concentration of the antibodies, in addition to the potential endogenous glutarylation (Figure 1, lower panels at the same intensity for all dots). In fact, at the optimal dilution, both the home-made and commercial anti(ε-glutaryl-lysine) antibodies, showed a minor reactivity to the non-glutarylated GDH, with such a reactivity of BSA being negligible. This corresponded to the known glutarylation of GDH which was detected also in the current study by our MS procedure, and there was no MS-detectable glutarylation of the BSA preparation [32]. Remarkably, both with the home-made and commercial antibodies, the reactivity of the non-glutarylated GDH to anti(ε-glutaryl-lysine) antibodies did not change after the acetylation or succinylation of GDH, but strongly increased after the enzyme glutarylation (Figure 2). Additionally, with BSA, only glutarylation, but not acetylation or succinylation, strongly increased its reactivity to both our and commercial anti(ε-glutaryl-lysine antibodies. This indicates the antibodies selectivity towards glutarylated lysine residues in proteins.

Although the specificity of the obtained antibodies towards ε-glutaryl-lysine residues of different proteins allows one to use them for the determination of glutarylated proteins in animal tissues, one should keep in mind that the degree of the protein glutarylation under physiological conditions is not comparable to that after the chemical modification by glutarylation agents [6]. Even for the most common acyl modification of lysine residues, i.e., acetylation, the extent of modification is far from 100% and depends on the physiological state and specific features of tissue metabolism [17,22]. Therefore, the antibody concentrations for the determination of purified proteins, artificially glutarylated with a high degree of modification, were not applicable for the analysis of protein glutarylation in the complex biological samples. In fact, Figure 3 shows that, despite the intense staining of glutarylated GDH and BSA, the used concentration of the anti-(ε-glutaryl-lysine) antibodies (0.016 µg/mL) hardly detected glutarylated proteins in extracts of liver mitochondria.

In this case, the determination of glutarylated proteins in the extract required a longer exposure and/or image contrasting (Figure 3a; lanes 8a and 17a which are presented separately). Nevertheless, this concentration of antibodies was enough to visualize the mitochondrial proteins after their artificial modification with glutaric anhydride (Figure 3a; gel lanes 9 and 18). At the same time, a comparison of the native and glutarylated extracts of the liver mitochondrial fraction showed that treatment with glutaric anhydride not only led to the appearance of multiple bands that were absent in the non-treated extract, but also increased the glutarylation of proteins that were relatively weakly glutarylated under native conditions (protein bands with molecular weights of 75 and 130 kDa, Figure 3). This experiment demonstrated both a low level of modification of potentially glutarylated proteins under native conditions and a rather high specificity of the obtained anti-(ε-glutaryl-lysine) antibodies towards this modification, regardless of the particular protein.

### 3.3. Characterization of Protein Glutarylation in the Rat Brain and Liver Homogenates

The obtained anti-(ε-glutaryl-lysine) antibodies were used to detect glutarylated proteins in the rat brain and liver homogenates. As can be seen from Figure 4, brain homogenates exhibited one major glutarylated protein band with a characteristic molecular weight of 75 kDa, and several minor ones. The pattern of protein glutarylation in the liver homogenates differed from that in the brain homogenates (Figure 4), with more glutarylated bands observed in the former than in the latter sample. Two of the main fractions of glutarylated proteins in the liver homogenates were characterized by the same molecular weights (75 and 130 kDa) as the glutarylated proteins determined in the mitochondrial extract (Figure 2). It should be noted that none of the identified fractions coincided in molecular weight with glutaryl-GDH (55 kDa) and glutaryl-BSA (69 kDa), the bands of which were present on the immunoblot of purified proteins (Figure 3) in the corresponding region between the 50 kDa and 75 kDa markers. The failure to detect glutarylated GDH in brain extracts from control rats was consistent with the absence of noticeable GDH glutarylation in the murine brain under physiological conditions [6]. Our mass spectrometric identification of rat GDH and serum albumin in brain homogenates also did not reveal their glutarylated fragments, although serum albumin was abundant in these samples: up to 60 peptides covering 65% of the serum albumin sequence were determined in the rat brain homogenates. This was consistent with a good separation of the protein bands corresponding to 69 kDa serum albumin and 75 kDa glutarylated protein (Figure 3 and Figure 4), providing additional evidence that antibody staining of the 75 kDa band in the rat brain and liver homogenates did not originate from the reaction with albumin.

Incubation of homogenates at 37 °C resulted in a redistribution of the intensity of several glutarylated protein bands. The observed changes indicated the possibility of endogenous proteolysis, despite the presence of protease inhibitors in the homogenates. Thus, after a 3 h incubation of the brain homogenates, the intensity of the low-molecular-weight band at the 20 kDa level increased, simultaneously with a decrease in the high-molecular-weight region (at 90 kDa level) of the 75 kDa protein peak (Figure 3c). Similarly, in the liver homogenate, the intensity of the protein band with a molecular weight of about 130 kDa decreased with a simultaneous increase in the intensity of the 75 kDa band. Thus, the possibility of endogenous proteolysis should be taken into account during the long-term incubations of the homogenates, which is often used in the studies of various factors potentially regulating protein glutarylation. Accompanying proteolytic degradation may result in an apparent increase or decrease in the intensities of the glutarylated protein bands due to the changes in the amount of protein in these bands.

### 3.4. Specific Patterns of Glutarylation, Succinylation and Acetylation in Biological Preparations

The degree of possible cross-interaction of the obtained antibodies with lysine residues modified by other acyl species was assessed by comparing the immunoblotting patterns of the same biological samples (rat brain homogenates) obtained using antibodies against lysine glutarylation, succinylation and acetylation (Figure 4).

As seen from Figure 5, the visualization of glutaryl-, succinyl- and acetyl-lysine residues of rat cerebral cortex proteins with appropriate antibodies revealed completely different patterns of the distribution of such residues over various proteins. For example, the main band of glutarylated proteins was characterized by a molecular weight of 75 kDa. The strongest staining of succinyllysine residues was observed in proteins having molecular weights of 32, 38 and 40 kDa. The main band of acetyl-lysine-containing proteins was 51 kDa. It should be noted that proteins of similar molecular weights as well as the same protein may undergo different types of acylation. Therefore, a staining of individual protein bands with different types of antibodies may be observed. For example, proteins in the region below 20 kDa, corresponding to histones undergoing various modifications affecting their electrophoretic mobility [3,11,12,15], may react with various anti-acyl-lysine antibodies. Nevertheless, the different overall patterns of specific acylation types, presented in Figure 5, confirmed the high specificity of the used antibodies towards the corresponding types of post-translational acylation of protein lysine residues.

## 4. Conclusions

The anti-(ε-glutaryl-lysine) antibodies obtained in this work after the two cycles of immunization and subsequent affinity purification on specific ligands enabled the specific detection of glutarylated proteins by immunoblotting of animal tissue preparations. These antibodies were reactive towards various glutarylated proteins. In addition to glutaryl-GDH and glutaryl-BSA used for immunization, the antibodies also interacted with the purified preparation of glutarylated MDH, as well as with glutarylated proteins of rat brain and liver homogenates, whose molecular weights were different from those of GDH and BSA. Cross-reactions with the non-modified proteins were mostly limited to endogenously glutarylated proteins and were not observed with the proteins after in vitro succinylation or acetylation. This selectivity provides a difference in the patterns of protein glutarylation, succinylation and acetylation in rat brain cortex homogenates, revealed by the antibodies of corresponding specificities. The comparative analysis of these patterns demonstrates the efficiency of the elaborated antibodies to detect specific types of protein acylations in complex biological samples. Immunoblotting of proteins of the rat brain and liver homogenates or mitochondria using the obtained anti-(ε-glutaryl-lysine) antibodies revealed a low level of proteins glutarylation in these tissues and subcellular fractions. In tissue homogenates, high-molecular-weight glutarylated proteins may undergo proteolytic fragmentation which must be taken into account when assessing and comparing glutarylation patterns.

## Figures and Tables

**Figure 1 biomolecules-11-01168-f001:**
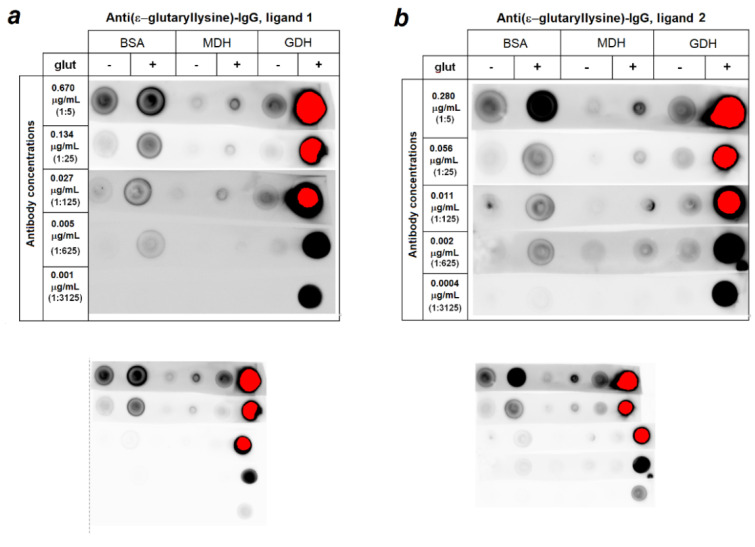
Immunodot titration results of affinity purified anti(ε-glutaryl-lysine) antibodies prepared from IgG fraction of antiserum obtained after the two immunization cycles. (**a**)—Titration of the antibodies purified from the ligand-1-comprising matrix; (**b**)—titration of antibodies purified from the ligand-2-comprising matrix. For better visualization of the results, different contrast was employed for the upper two and lower three rows. Lower figures present the visualization of all rows at the same brightness. Red-colored spots correspond to an extremely high luminescence.

**Figure 2 biomolecules-11-01168-f002:**
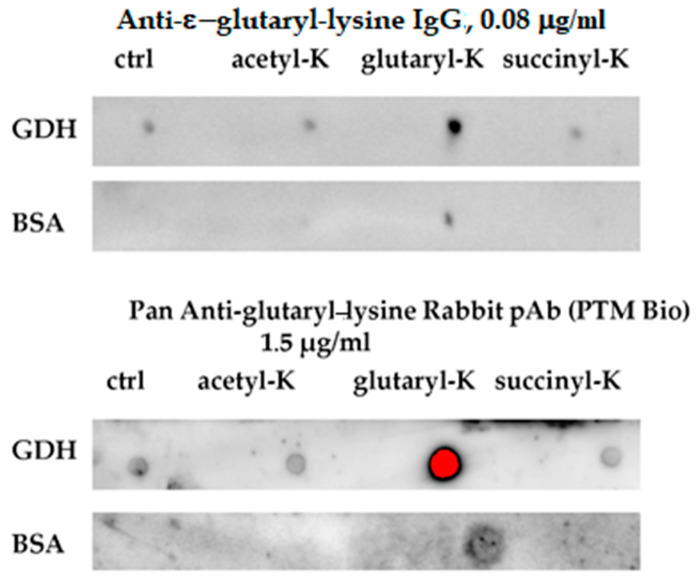
Comparison of specificity and selectivity of the home-made and commercial anti-(ε-glutaryl-lysine) antibodies, using purified proteins with artificially modified lysine residues (see details in Materials and Methods). An amount of 1.5 μg/mL concentration of commercial antibodies represents the one recommended for immunoblotting by the manufacturer.

**Figure 3 biomolecules-11-01168-f003:**
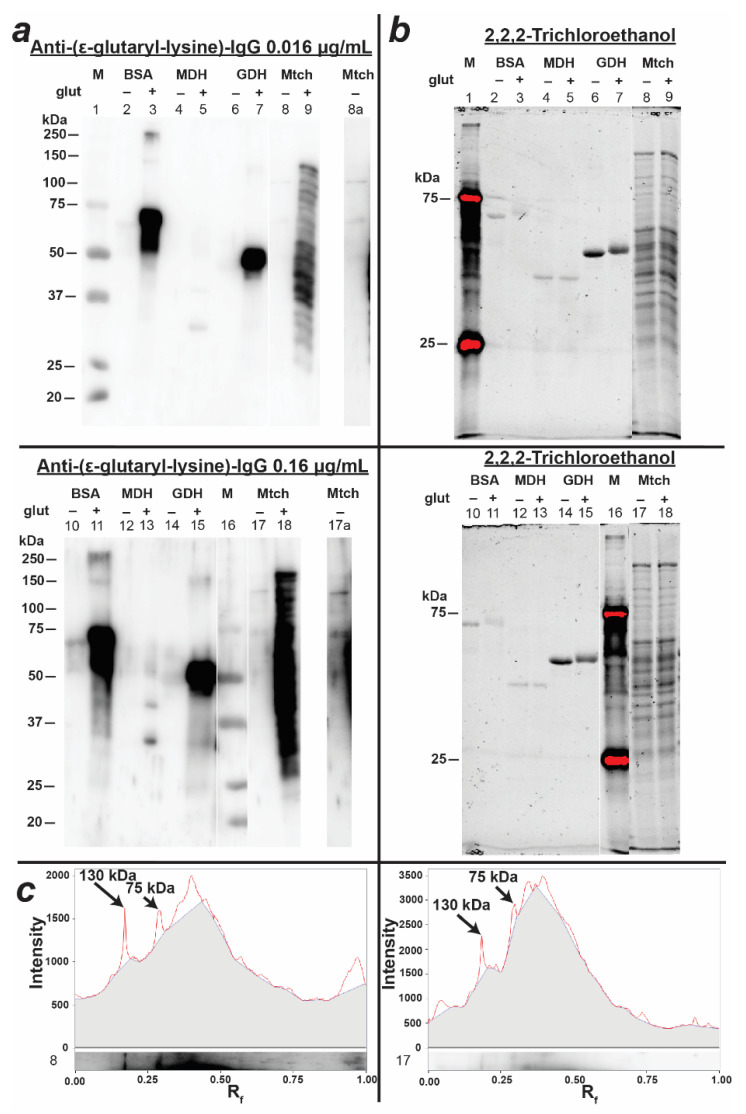
Immunovisualization of glutarylation of purified proteins and proteins of the extract of liver mitochondria, prepared from male rats aged 10–12 weeks. The results of incubation of PVDF membranes with anti-(ε-glutaryl-lysine) antibodies at 0.016 μg/mL and 0.16 μg/mL concentrations (**a**) and total protein in the corresponding gels, assessed by 2,2,2-trichloroethanol staining (**b**). Lane numbers correspond to the following samples: 1, 16—protein markers (M) with known molecular weights indicated to the left of the blots; 2, 10—untreated bovine serum albumin (BSA); 3, 11—glutarylated BSA; 4, 13—untreated malate dehydrogenase (MDH); 5, 13—glutarylated MDH; 6, 14—untreated glutamate dehydrogenase (GDH); 7, 15—glutarylated GDH; 8, 17—untreated extract of liver mitochondria (Mtch); 9, 18—glutarylated mitochondrial liver extract; 8a, 17a—contrast images of lanes 8, 17 for better visualization of bands with molecular weights of 75 and 130 kDa. (**c**) Densitograms of the untreated extract of liver mitochondria (lanes 8 and 17); the arrows show the peaks corresponding to the proteins mentioned in the text with molecular weights of 75 and 130 kDa; grey color corresponds to the background staining from adjacent lanes (9 and 18).

**Figure 4 biomolecules-11-01168-f004:**
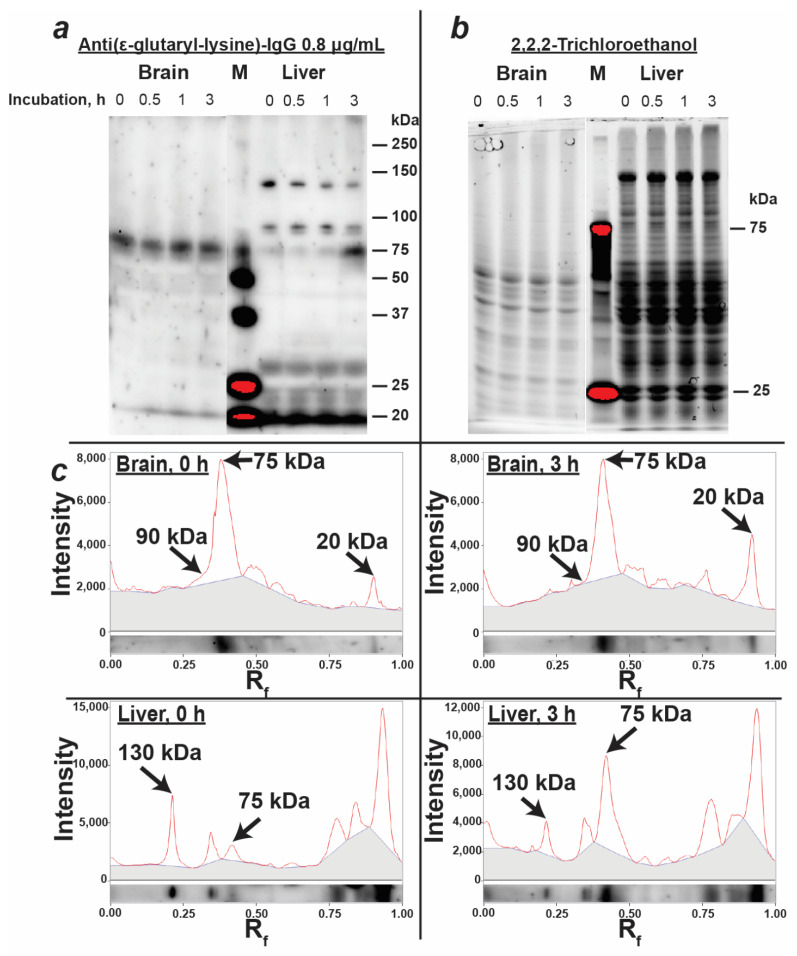
Detection of glutarylated proteins in homogenates of the cerebral cortex and liver of male rats aged 8–9 weeks by immunoblotting using anti-(ε-glutaryl-lysine) antibodies. (**a**) Immunovisualization of glutarylated proteins of the cerebral cortex and liver homogenates on the PVDF membrane; (**b**) visualization of proteins of the homogenates in SDS-PAGE before transfer onto the membrane; (**c**) densitograms of bands of glutarylated proteins of the rat brain and liver homogenates without incubation and after 3 h of incubation under conditions specified in Materials and Methods; the arrows show the peaks corresponding to the proteins mentioned in the text with a molecular weight of 75, 90 and 130 kDa and a low molecular weight band (20 kDa); grey color corresponds to the background staining contributed by adjacent lanes. M—protein markers with known molecular weights indicated at the right side of the blots.

**Figure 5 biomolecules-11-01168-f005:**
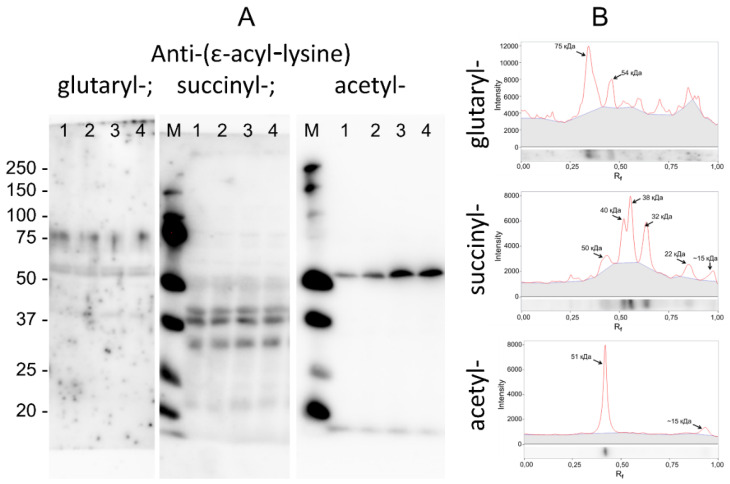
Visualization of glutarylation, succinylation and acetylation of proteins of the brain cortex of the male rats aged 10–12 weeks. Proteins were separated by SDS-PAGE and visualized on the membrane using the obtained antibodies against ε-glutaryl-lysine (0.8 μg/mL) and commercial antibodies against ε-succinyl-lysine (0.5 μg/mL) and ε-acetyl-lysine (0.29 μg/mL). (**A**) Immunoblotting of the cerebral cortex samples from male rats; 1–4—different animals; M—protein markers with known molecular weights indicated to the left of the blots. (**B**) Densitograms of the results of immunoblotting with automatically determined molecular weights of the major protein bands.

**Table 1 biomolecules-11-01168-t001:** Glutarylated lysine residues of the GDH peptides determined by mass spectrometry analysis of the purified enzyme after modification with 0.16 mM glutaric anhydride (this work) and 0.5 mM glutaryl-CoA [6]. * Lysine residue preceded by glycine residue.

Glutarylated Lys Residue of GDH	Modification with 0.16 mM Glutaric Anhydride	Modification with 0.5 mM Glutaryl-CoA [6]
K90	-	+
K171	+	+
K183	+	+
K187	+	-
K191	+	+
K200	-	+
K352	-	+
K365	-	+
K386	-	+
K390	+	-
K457	+	-
K477	+	+
K480 *	+	+
K503	+	+
K527	-	+

## Data Availability

All primary data supporting the results are available directly from the authors upon request.

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
