# Peer review of "Preparation of Affinity Purified Antibodies against ε-Glutaryl-Lysine Residues in Proteins for Investigation of Glutarylated Proteins in Animal Tissues"

_biomolecules, 2021, doi:10.3390/biom11081168_

Round 1

Reviewer 1 Report

Protein modifications play important roles in many essential biological processes. One way for protein modification study is to identify and map the modified proteins and residues by antibodies. In this manuscript “Preparation of affinity-purified antibodies against e-glutaryl-lysine residues in proteins for investigation of glutarylated proteins in animal tissues”, Kolesanova et al presented a method for pan anti-glutarylated protein antibody using sequential immunization with two artificially glutarylated proteins and glutaryl-lysine-containing sorbents. The immunization with two different glutarylated protein increase PTM specificity of antibody by reducing its dependence on carrier proteins. The purified antibody could be used to detect tissue and subcellular protein glutarylation. The specific patterns of lysine glutarylation based on the purified antibody were different from succinylation and acetylation. However, I have the following major concerns.

  1. To test the efficiency of purified anti-Kglu antibody, in Figure 1a, the authors used unmodified and glutaric anhydride-treated proteins. However, there was still a signal detected from those unmodified proteins. The author claimed that it could be caused by their initial glutarylation level. To further confirm the specificity of the anti-Kglu antibody, the author can perform the dot blotting using unmodified and glutarylated peptides corresponding to those proteins.
  2. Pan anti-Kglu antibodies are commercially available, including PTM-1151 (PTM Biolabs) and #26101 (Cell signaling). The comparison between the purified and commercially available anti-Kglu antibodies should be performed to show the high efficiency and specificity of the purified antibody.
  3. The titer of antibody against glutaryl-BSA after the second cycle of immunization increase significantly from 1:3200 to 1: 102400. How about the antibody titer against glutaryl-GDH or BSA after affinity purification using lysine glutarylated peptides?
  4. To test the cross-interaction of purified anti-Kglu antibody with other acylations, the author compared protein patterns using antibodies against different acylation. A better approach to show the specificity of the purified antibody is to perform immunoblotting analysis use the same peptide or protein with different acylation (glutarylation, succinylation, malonylation, HMGylation, and acetylation).
  5. The protein patterns in Figure 3a were not consistent with those in Figure 4a even though using the same antibody and same tissue.

Reviewer 2 Report

In this paper, Kolesanova et al. describe the isolation of polyclonal antibodies, targeting ɛ-glutaryl-lysine residues in proteins, following rabbit immunization. The authors used two different glutarylated proteins for immunization and specific antibodies were later isolated from the rabbit sera using affinity purification. The authors exhibit the use of these antibodies for identification of glutarylated proteins in various tissues and show that these could be distinguished from other lysine modifications.

The paper is coherently displayed and well written. However, some minor points should be addressed.

  1. The authors state that the goal of this work was “…to obtain antibodies that specifically interact with ε-glutaryllysine residues for the determination of glutarylated proteins in animal tissues” (lines 84-85). However, the purpose of identifying such proteins in animal tissues is not clear enough. This point should be addressed more clearly in the introduction.
  2. Table 1 presents the lysine residues of GDH modified by the reagent used in this study, compared to another agent used in a different study. The authors state that this comparison “…showed a significant, but incomplete, overlap of the enzyme lysine residues modified by both agents” (lines 240-241). However, only 6 out of 15 residues presented in the table, overlap. This is not significant. Also, the glutaryl-CoA agent seems to be much more efficient. Why wasn’t it used for the modifications performed?
  3. Although the specificity of the purified antibodies towards glutarylated residues is obvious from Figure 1, it can not be ruled out that some of the antibodies in the polyclonal preparation recognize the protein itself that was used for immunization (GDH or BSA), which can explain the reaction towards the unmodified proteins, the much weaker reaction towards modified MDH and even the differences in intensity between modified GDH and BSA. Authors should relate to this point.
  4. In Figure 2, the use of two antibodies concentrations to visualize native glutarylated proteins is presented. Antibodies in both of these concentrations fail to detect glutarylation in both purified proteins and liver mitochondria (lines 331-333, Figure 2). Since the purpose of this study is to detect these modifications in tissues, and not in treated proteins, these results are somewhat problematic. why weren’t higher concentrations of antibodies used?
  5. In figure 3, antibodies in lower concentrations (0.08 µg/ml compared to 0.16 µg/ml, presented in figure 2) do detect glutarylated proteins in rat brain and liver. How can the authors explain these differences between the experiments?
  6. Figure 4: the concentrations of the antibodies used are not mentioned. Please add.
  7. Lines 450-452: the authors state that “No cross-reactions were observed between the anti-(ε-glutaryllysine) antibodies and succinylated and acetylated proteins”, however, cross-reaction was not measured. The patterns of different modifications in biological preparations were compared, using different antibodies for each modification (Figure 4). In order to determine that there is no cross-reactivity, the experiment should be performed using the anti-(ε-glutaryllysine) antibodies against the succinylated and acetylated proteins.

Round 2

Reviewer 1 Report

The authors largely addressed my concerns and I support the publication of this study with minor revision described below.

  1. The MS2 spectrum should be provided for endogenous GDH peptide containing K503 glutarylation and the corresponding unmodified one.
  2. Line 338: “…although this modification was not detectable...”
  3. The font format is not consistent in Figure 2. i.e. ug/ml.

Author Response

  1. The MS2 spectrum should be provided for endogenous GDH peptide containing K503 glutarylation and the corresponding unmodified one.

A Supplement Figure 1S comprising MS2 spectra of GDH peptide fragment with Lys503 residue, either glutarylated (A) or non-glutarylated (B), is added to the manuscript. 

2. Line 338: “…although this modification was not detectable...”

Thank you for the note. The typing error (no spacing before “was”) is corrected.

3. The font format is not consistent in Figure 2. i.e. ug/ml.

Thank you for the note. The font is corrected.